# Epstein–Barr Virus History and Pathogenesis

**DOI:** 10.3390/v15030714

**Published:** 2023-03-09

**Authors:** Hui Yu, Erle S. Robertson

**Affiliations:** 1Department of Hematology, The Affiliated Hospital of Nanjing University of Chinese Medicine, Jiangsu Province Hospital of Chinese Medicine, Nanjing 210029, China; 2Departments of Otorhinolaryngology-Head and Neck Surgery, and Microbiology, The Tumor Virology Program, Abramson Cancer Center, Perelman School of Medicine at the University of Pennsylvania, Philadelphia, PA 19104, USA

**Keywords:** Epstein-Barr virus, pathogenesis, cancer induction, lymphoma, epithelial carcinoma

## Abstract

Epstein–Barr virus (EBV) is the first identified human oncogenic virus that can establish asymptomatic life-long persistence. It is associated with a large spectrum of diseases, including benign diseases, a number of lymphoid malignancies, and epithelial cancers. EBV can also transform quiescent B lymphocytes into lymphoblastoid cell lines (LCLs) in vitro. Although EBV molecular biology and EBV-related diseases have been continuously investigated for nearly 60 years, the mechanism of viral-mediated transformation, as well as the precise role of EBV in promoting these diseases, remain a major challenge yet to be completely explored. This review will highlight the history of EBV and current advances in EBV-associated diseases, focusing on how this virus provides a paradigm for exploiting the many insights identified through interplay between EBV and its host during oncogenesis, and other related non-malignant disorders.

## 1. Introduction

Epstein–Barr virus (EBV or human herpesvirus 4 (HHV-4)), is a member of the γ-herpesviruses family. It is the first human tumorigenic virus to be characterized. It is one of the most successful viruses, infecting up to 95% of the world’s adult human population and sustaining a life-long asymptomatic infection of the B-lymphocyte pool [1,2,3,4]. Natural EBV infection is strikingly ubiquitous in humans [2,5]. Studies have elucidated its role as a causative factor in the development of a diverse spectrum of diseases, including benign diseases (infectious mononucleosis (IM)) [2], oral diseases [6], diseases related to functional abnormalities of immunity, multiple sclerosis (MS) [7], systemic autoimmune diseases (SADs) [8], various malignancies (hematological malignances, epithelial cancers) [3,9], and EBV-associated hemophagocytic lymphohistiocytosis (EBV-HLH).

It is estimated that more than 250,000 cases of cancer every year are induced by EBV, and that approximately 2% of all cancer deaths are due to EBV-attributed malignancies [10]. Improved understanding of pathogenic mechanisms, which drives the many aforementioned diseases, provides clues as to potential opportunities for recognizing and identifying early and effective treatments at different stages in the development of EBV-associated diseases. In addition, the more recent utility of immunotherapy to target EBV-infected cells can potentially overcome viral-mediated immune evasion. Over the years, many vaccine strategies and therapeutic agents have been developed to target the EBV gene products, as well as to promote immune surveillance.

## 2. EBV History

EBV was discovered by electron microscopy of cells cultured from fresh African Burkitt’s lymphoma (BL) biopsy after outgrowth of the cells in culture in the laboratory of Dr. Anthony Epstein in 1964 [11]. There was almost immediate speculation that the virus might be involved in the development of tumorigenesis. Since then, EBV has been an object of intense interest. In 1966, studies showed that there were high EBV antibody levels in BL and nasopharyngeal carcinomas (NPCs) patients [12], as well as in healthy donors [13]. EBV was shown to be the causative agent of IM in 1968 [14]. The presence of EBV DNA was reported in BL cells, as well as extracts from anaplastic NPC samples, which were derived from patient biopsies in 1970 [15]. It was then further confirmed that EBV-specific nucleic acids were present within NPC cells [16]. In the 1980s, EBV was found to be associated with non-Hodgkin’s lymphoma and oral hairy leukoplakia in patients with acquired immunodeficiency syndrome (AIDS) [17,18]. Since then, EBV DNA has been found in tissues from a wide range of different cancer types. Those include hematological disorders, such as post-transplant lymphoproliferative disorders (PTLDs) [19]; Hodgkin’s lymphoma [20]; T-cell lymphoma [21]; NK cell leukemia and other T-cell, NKT-cell and NK-cell lymphoproliferative diseases [22,23]; and certain epithelial neoplasms, including lymphoepithelioma-like carcinomas occurring in a variety of organs, such as gastric adenocarcinoma [24], breast cancer [25], and liver carcinoma [26]. More recently, several SADs and MS have been shown to correlate with chronically reactivated EBV infection and dysfunctional immune control of the virus [8]. However, the detailed contributions of EBV to the pathogenesis of the majority of the aforementioned diseases remain to be explored. Therefore, a thorough examination of the infection, reactivation, and cell transformation induced by EBV will continue to provide more details of its contribution in driving pathogenesis, which will eventually lead to the development of therapeutic targets, agents, and strategies for clinical interventions.

## 3. Biology and the Potent Transforming Ability of Oncogenic EBV

The stages of the EBV life cycle include the primary infection, establishment of latency, and reactivation or lytic stage to produce new virions, and depend on the interplay between the virus and the host immune system. The fact that EBV persists for the lifetime of the host indicates that it has evolved successful strategies to obstruct the host immunity. Typically, the majority of primary EBV infections are asymptomatic, which occurs during infancy and early childhood in areas where general hygiene conditions are less strictly enforced. Teens and young adults who have not been exposed to EBV early in life are prone to develop IM, which is also called “kissing disease”, as the major transmission route from the carrier to the naive host occurs orally through direct contact with virus-containing saliva, such as with kissing. It can also be transferred via semen during sexual contact, organ transplantation, and blood transfusion [6]. Following primary infection, EBV displays latent infection in the circulating B-cell pool of peripheral blood, minimizes viral production to evade the immune system, and persists in the host with minimal impact. Acute infection and periodic reactivation, on the other hand, allows the infectious progeny virus to spread to new hosts.

Oropharyngeal epithelial cells and B cells are the typical cell types that have shown strong tropism for EBV [8]. The question of the original cell type (either oropharyngeal epithelial cells or B cells) to first get infected by EBV during salivary transmission remains a debate. It is speculated that EBV first infects the oropharyngeal epithelial cells, where it replicates and releases virions. The newly produced virions then infect the co-localized B cells in the lympho–epithelial structures of Waldeyer’s ring [27]. However, there is an opposite point of view assuming that EBV is transmitted via the saliva, enters the tonsillar crypts, crosses the thin layer of epithelium overlying the bed of lymphocytes below, and infects naive B cells residing in the follicular mantle by a yet to be determined mechanism [28]. This point of view was supported by a prospective study that showed that EBV viral genomes are detected by nested PCR at low levels in peripheral blood about three weeks before any IM symptoms, suggesting that B cells are the major reservoir of the virus before epithelial amplification in the oral cavity [29]. Human primary B lymphocytes, as the major reservoir for EBV, are infected through interaction between the major viral outer envelope glycoprotein gp350/220 and the CD21 receptor on the surface of B cells, together with gp42, which is essential for penetration of B cells by forming a complex along with the histocompatibility complex (MHC)-II [30]. EBV can shuttle between different cell types under certain circumstances, mainly B cells and epithelial cells. However, other lymphocytes, such as T cells, NK-lineage cells, monocytes, smooth muscle cells, follicular dendritic cells (DC), and even neurons, can also become infected, as well [31,32,33,34]. Oropharyngeal epithelial cells, unlike B lymphocytes, are permissive for viral replication [35,36], and it was demonstrated that EBV binds much less efficiently to epithelial cells than B cells. The mechanism of how EBV enters into primary B cells and nasopharyngeal epithelial cells has been largely elucidated. However, further understanding of EBV entry into other different cell types is still somewhat incomplete, requiring further examination (Figure 1).

EBV initiates a distinct dual-life cycle, either latent (nonproductive) infection or lytic (productive) replication. The virus undergoes predominantly lytic replication in epithelial cells in vitro and establishes lifelong latency in circulating memory B lymphocytes, but can reactivate periodically from latency. As a result, EBV can shuttle between different cell types, mainly B cells and epithelial cells. EBV can also have chronic active recurrence or reactivate based on specific triggers that results in switching between the latent and lytic life cycle. Moreover, during latent infection, only a limited number of genes are expressed, which are required for maintenance of the viral genome (as an episome in the nucleus), evasion of the host immune system surveillance, cell growth, and proliferation.

The B-lymphotropic nature of EBV is evidenced by the potent transforming ability of the virus to immortalize normal resting B lymphocytes in vitro, converting them into permanently growing lymphoblastoid cell lines (LCLs) [39,40]. EBV remains the most efficient transforming agent, which rapidly immortalizes B cells during in vitro infection. The complete sequence of an EBV strain (the B95-8 strain) was first reported in 1984 [41], and 84 different EBV strains have been isolated to date [38]. There are two types of EBV subtypes, type 1 and type 2; the major identified differences between these two types are the sequences of the latent infection cycle nuclear antigen genes EBNA2, EBNA-LP, and EBNA3 [42,43] and the small, non-polyadenylated RNAs EBER1 and EBER2 [44]. Type 1 strains are more prevalent worldwide and have greater transforming potential [45]. There are four patterns of EBV latency [46,47], which were defined based on the pattern of EBV proteins and small RNAs expressed in in vitro infections and in EBV-induced tumors [48] (Table 1). The EBV genome within LCLs usually expresses all latent genes, which is also known as the latency Ⅲ (growth program), which includes six Epstein–Barr nuclear antigens (EBNA1, EBNA2, EBNA3A, EBNA3B, EBNA3C, and EBNA leading protein (EBNA-LP)), three latent membrane proteins (LMP1, LMP2A, and LMP2B), two small nonpolyadenylated RNAs (EBER-1 and EBER-2), and transcripts from the BamHI-A region (BARTs, also known as miRNAs) [49]. The latency Ⅲ program also occurs in iatrogenic immunodeficiency lymphoma (i.e., post-transplant lymphoproliferative disease (PTLD)), primary CNS lymphoma (HIV-associated), NHLs with primary immune disorders, DLBCL associated with chronic inflammation, and EBV-associated T- and NK-cell lymphomas [32]. Latency Ⅰ is described as a program where EBNA1 is the only viral protein expressed, and is seen in immunocompetent/immunocompromised BL; primary effusion lymphoma (PEL); and plasmablastic lymphoma, oral type, in immunodeficient HIV patients [50]. Latency Ⅱ (also referred to as the default program), in which EBNA1 and LMPs are expressed, is typically displayed in Hodgkin’s lymphoma; EBV-positive DLBCL; lymphomatoid granulomatosis; angioimmunoblastic T-cell lymphoma; extranodal NK/T-cell lymphoma, nasal type; aggressive NK-cell leukemia; NPC; and gastric carcinoma (GC). Latency 0, in which none of the EBV antigens are expressed, avoids immune recognition and is typically observed in circulating memory B cells in healthy persons. The expression of EBERs, as well as miRNAs, is present in all forms of latency. Furthermore, it has been documented that in the normal course of EBV infection in healthy individuals, even latency Ⅲ is very transiently seen. However, owing to fluctuations in immune pressures, these different forms of latency may exist [3,51]. Although the different types of latency programs predominate in different EBV-driven tumors, lytic viral replication has also been shown to have pathogenic importance [52,53,54]. Considering the relatively low incidence of EBV-associated malignances when compared to the high proportion of EBV infection in the worldwide population, a detailed understanding of the mechanism of pathogenesis underlying its contribution to the different associated malignancies needs further investigation.

## 4. Pathogenesis Associated with EBV Latent and Lytic Infection

### 4.1. EBV Latent Infection. Contribution of EBV Latent Antigens to the Oncogenic Phenotype Remains the Subject of Intense Study

#### 4.1.1. The EBV-Encoded Nuclear Antigens (Summarized in Inset 1)

EBV-encoded nuclear antigens (EBNAs) can influence both viral and cellular transcription. EBNA1 is the only known EBV protein to be consistently expressed in all EBV-related cancers. Its main identified function is to ensure the replication of the double-stranded DNA (dsDNA) EBV genome as cells multiply [55,56]. It can enhance cell survival by inhibiting apoptosis [57], induce genomic instability via the production of reactive oxygen species (ROS) [58], and induce B-cell neoplasia in transgenic mice [59]. EBNA1 also contributes to transcriptional regulation of the EBNAs (including EBNA1 itself) and LMP1 by interacting with certain viral promoters [4]. The main function of the Gly-Ala repeat sequence domain of EBNA1 serves to prevent the protein from proteasomal degradation so as to avoid presentation as peptides to cytotoxic T lymphocytes (CTLs) [60,61]. EBNA1 can also counteract the stabilization of the host-encoded p53/TP53 and MDM2 by host ubiquitin-specific protease USP7, thereby decreasing apoptosis and increasing host cell survival [62]. High-affinity molecular mimicry between EBNA1 and the central nervous system protein glial cell adhesion molecule (GlialCAM) in MS was recently demonstrated. Meanwhile, structural and in vivo functional evidence for their relevance was also illustrated [63]. Taken together, the function of EBNA1 is clearly not restricted to simply maintaining the viral genome in EBV-infected cells.

EBNA2 is an EBV-encoded nuclear transcription factor that plays an essential role in the transformation process in vitro. It is the essential EBV transactivator that is required for activating cellular and viral genes. It can directly activate the important viral latent genes, which include EBNAs and LMPs genes, along with other cellular genes, such as MYC and CD23 [64,65,66]. EBNA2 was also highlighted in having a critical role in rewiring gene regulatory programs of the host through rearrangement of the chromatin landscape, which suggests that these interactions are potential components of genetic mechanisms that can influence the risk of multiple autoimmune diseases [67,68]. EBNA2 can functionally replace the intracellular region of Notch to interact with RBP-Jk to derepress the transcriptional activity at its responsive binding sites, and has been implicated in the development of EBV-associated T-cell tumors in humans [69,70].

EBNA-LP functions as a transcriptional coactivator by cooperating with EBNA2 in an RBP-Jk-mediated large multi-protein complex in the transcriptional activation of responsive cellular and viral promoters. These activities have been shown to be modulated by the EBNA3 family of proteins, which results in modulation of transcription activation through its interaction with cellular and viral transcription activator complexes [4,71,72]. EBNA-LP may also play a role in evading immune surveillance by suppressing the innate cell response to viral DNA, which allows transcription of the EBV genome and survival of EBV-infected naive B cells [73].

The EBNA3 family of proteins comprises EBNA3A, EBNA3B, and EBNA3C. Both EBNA3A and EBNA3C are considered oncogenic and are required for efficient transformation of B cells [74,75]. EBNA3A and EBNA3C can mediate the inhibition of various cyclin-dependent kinase inhibitors [76,77] and potentiate EBV-mediated B-cell transformation and subsequent viral persistence by inhibiting the apoptotic response to viral infection and cell transformation [50,78,79]. This includes inhibition of the BIM gene (a member of the pro-apoptotic B-lymphoma-2 gene (BCL-2) family), further promoting virus-induced proliferation and tumorigenesis [80]. EBNA3A and EBNA3C help establish a long-term latency and subsequent lymphoma development by blocking B-cell differentiation to plasma cell phenotype through transcriptional activation of the cyclin-dependent kinase inhibitor p18^INK4c^ and the master transcriptional regulator of plasma cell differentiation, B lymphocyte-induced maturation protein-1 (BLIMP-1) [81]. In contrast, EBNA3B functions distinctly from EBNA3A and EBNA3C as a tumor suppressor, whose inactivation promotes immune escape and EBV-driven lymphomagenesis [82,83].



**Inset 1.** Summary of the functions associated with latent EBV nuclear antigens.

#### 4.1.2. EBV-Encoded Latent Membrane Antigens (Summarized in Inset 2)

LMP1 is the main transforming antigen of EBV; it functions like a constitutively active CD40 in a ligand-independent manner [84]. LMP1 has a large number of pleiotropic effects; it can block p53-mediated apoptosis, by stimulating the expression of the zinc finger protein A20 (A20) gene that promotes cell proliferation, and regulate the inflammatory response [85,86]. LMP1 mimics the activated tumor necrosis factor receptor (TNFR) superfamily member CD40 receptor and recruits important intracellular adaptor molecules, TNFR-associated factor (TRAF) family members TRAF1, TRAF2, and TRAF3, to activate nuclear factor κB (NF-κB)-inducing kinase (NIK) and inhibitory kappa B kinases (IKKs), which phosphorylate IKKα and further promote cell survival by inhibiting apoptosis [84,87]. LMP1 also promotes B-cell proliferation by inducing a STAT1-dependent IFN-γ secretion [88] and upregulates anti-apoptotic proteins (BCL-2, myeloid cell leukemia protein-1, BCL2A1 homology, A20) by inducing the expression of cell surface adhesion molecules CD23, CD40, intercellular adhesion molecule-1, and lymphocyte function-related antigen-3 (LFA-3) to inhibit apoptosis and promote tumorigenesis [89,90]. The oncogenic ability of LMP-1 can also be attributed to its ability to induce the activation and secretion of different matrix metalloproteinases, suggesting an important role for this oncoprotein in both the angiogenic and metastatic process during the onset and development of EBV-associated tumors [91]. LMP1 can utilize the PI3K–AKT–mTOR pathway for inducing CD137 expression to support growth of the Hodgkin and Reed/Sternberg (HRS) cells and escape from immune surveillance [92]; it can also promote tumor immune escape by upregulating tumor cell PD-L1 through the NF–κB pathway in extranodal natural killer/T-cell lymphoma (ENKTL) [93].

The LMP2 proteins, LMP2A and LMP2B, first identified in 1990 [94], are not essential for EBV-induced transformation in vitro [95]. LMP2A can drive the proliferation and survival of B cells in a BCR-deficient context, which indicates that LMP2A is a mimic of BCR [96,97]. LMP2A mimics a subset of BCR signaling events, including tyrosine phosphorylation of the kinase SYK, the calcium initiation complex consisting of BLNK, BTK, and PLCγ2, and its downstream transcription factor NFAT [98]. LMP2A affects apoptosis and cell-cycle checkpoints by dysregulating the expression of apoptotic regulators, such as BCl-xL and the tumor suppressor retinoblastoma-associated protein 1 (RB1) [98]. LMP2A can also transform epithelial cells and enhance their adhesion and motility via the PI3K–Akt pathway [99]. Furthermore, LMP2A attenuates signaling from type I and type II interferon receptors (IFNRs), which circumvents the innate immune response, an effect that is generally exploited in persistent virus infections, but which can, in the context of oncogenic viruses, such as EBV, contribute to tumor development by stimulating cell growth and limiting cellular responses to virus-infected cells [100]. The detailed functions of LMP2B remain obscure, although some studies have suggested that it serves to modulate the effects of LMP2A on BCR function, thereby rendering latently infected B cells susceptible to lytic re-activation [101,102].



**Inset 2.** Summary of the functions associated with EBV-encoded latent membrane antigens.

#### 4.1.3. EBV-Encoded Functional Small Non-Coding RNAs

Two small non-coding RNAs, EBER1 and EBER2, are expressed in all forms of latency. However, the precise function of EBERs remains to be elucidated. EBERs are not essential for EBV-induced transformation of primary B lymphocytes in vitro [4]. They can increase tumorigenicity, promote cell survival, induce interleukin-10 (IL-10) expression in BL cell lines [103,104,105], and significantly enhance the tumorigenicity of EBV-negative BL cells in SCID mice [106]. The *Bam*H1A right frame 1 (BARF1), a transcript generated from the *Bam*H1A region, shares limited homology with the human colony-stimulating factor 1 receptor (*CSF1R*), also called the *FMS* proto-oncogene, and displays oncogenic activity when expressed in rodent fibroblasts and primary monkey epithelial cells [107,108]; it is also capable of activating anti-apoptotic Bcl-2 expression [108]. It is originally identified as an early antigen expressed on induction of the EBV lytic phase, but has been shown to be expressed as a latent protein in EBV-associated NPC and GC [109,110,111].

### 4.2. EBV Lytic Genes and Their Associated Functions

Latency occurs predominantly in B lymphocytes; however, lytic infection can occur in both lymphocytes and epithelial cells. While EBV tumors are comprised of cells in the different latency programs, multiple studies have pointed out that lytic infection, which leads to high viral load, is a cancer determinant in EBV-induced malignances [54]. It has become increasingly clear that, in certain circumstances, the lytic cycle contributes to EBV-induced oncogenesis [112,113]. Lytically infected cells could release growth factors and immunosuppressive cytokines capable of signaling through many different signaling pathways, leading to tumor growth enhancement [53]. Besides the contribution of the lytic cycle itself to the development of EBV-induced cancers, numerous studies have showed that specific lytic proteins can promote oncogenesis [114,115,116]. The two EBV immediate early transcriptional activators, *BZLF1* and *BRLF1*, can synergistically induce expression of multiple early lytic genes involved in viral DNA amplification, late gene expression, and virion production [38,117]. Expression of *BZLF1* (the latent to lytic infection switch protein) is the beginning of lytic infection, which has been found in BL, NPC, and GC samples [118,119,120]. *BZLF1* can also activate IL-8 and IL-10 expression, leading to the enhancement of cell growth and survival [121]. Another two EBV-encoded immediate early genes, *BHRF1* and *BALF1*, which were found to be turned off once latent infection is established, are homologues of the cellular, anti-apoptotic gene Bcl2, also named viral Bcl-2 proteins (vBcl-2s) [122,123,124]. Genetic inactivation of both these two vBcl-2 genes disabled EBV’s ability to transform primary resting B lymphocytes and resulted in immediate apoptosis of cells [124], which implied that there is a critical role for the vBcl-2s in enabling cell survival during the early stages of the transformation process. *BILF1* encodes a constitutively active G protein-coupled receptor, which is capable of modulating various intracellular signaling pathways, including the NF-κB-mediated signaling pathway [125].

Several EBV lytic genes are also known to induce genome instability. These include *BZLF1* (impairs accumulation of host DNA damage proteins) [126], *BGLF4* (induces the DNA damage response and chromatin remodeling through TIP60) [127], *BGLF5* (mediates DNA damage and repression of DNA repair, subsequently increasing microsatellite instability (MSI) and genetic mutations) [128], *BNRF1* (induces centrosome amplification) [129], *BPLF1* (interferes with DNA repair through its effects on PCNA and Pol η) [130], and *BKRF4* (binds histones and interferes with double-stranded DNA break repair) [131].

In addition, several lytic genes have been shown to contribute to immune evasion, including the innate and adaptive immune response. For example, *BPLF1* evades the host immune system through interference with the Toll-like receptor signaling pathway [132]; *BZLF1* downregulates MHC class Ⅱ genes [121]; *BNLF2a* inhibits the transporter of antigen processing [133]; *BILF1* induces MHC class I internalization and degradation [134]; *BZLF1* and *BNLF2a* synergistically inhibit the processing and presentation of cytotoxic T-cell epitopes [135]; *BGLF5* mediates host shutoff [136]; and *LF2* antagonizes type I interferon signaling leading in immune evasion [137]. Understanding the relationship between the EBV virus with its host cell and the interaction between EBV latent with lytic antigens in the context of associated cancers will continue to be an important topic for future studies.

## 5. EBV-Associated Malignances

### 5.1. EBV-Associated Lymphomas

#### 5.1.1. EBV-Associated B-Cell Lymphomas

##### Burkitt’s Lymphoma

Burkitt lymphoma (BL) is a highly aggressive mature B-cell neoplasm and includes the historically recognized three subtypes, which are endemic BL, sporadic BL, and immunodeficiency-associated BL. Endemic BL, with the EBV genome present in >95% of the neoplastic cells, occurs in sub-Saharan Africa and regions endemic for malaria [138]. All BLs have a characteristic chromosome translocation of the *MYC* gene to one of the *Ig* loci, which juxtaposes *MYC* to the downstream of *Ig* heavy or light chain enhancer, resulting in enhanced and deregulated C-MYC expression [138]. The translocations are thought to occur in the lymph node germinal center, where *Ig* gene modifications associated with class switching and somatic hypermutation normally happen in B cells [9]. In the context of endemic BLs, EBV and malaria induce activation-induced cytidine deaminase (AID) activity in the lymph node and eventually initiate *Ig*/*MYC* translocations [138]. It was also shown that EBNA2 can activate *MYC* and consequently increase the sensitivity of upstream enhancer regions to AID activity [139]. The fact that EBV, together with malaria, can increase the frequency of BLs is attributed to malaria-related higher EBV load, higher levels of AID, and the production of an increased number of B cells in response to malaria in the germinal center, and lower surveillance of EBV due to immunodeficiency [9]. Emerging evidence suggests a dual mechanism of BL pathogenesis: virus-driven versus mutational burden, which depends on the EBV status, regardless of the epidemiologic context and geographic location. Thus, based on recent insights into BL biology, it is recommended by the 5th edition of the WHO classification of hematolymphoid tumors (WHO-HAEM5) to classify BL into two subtypes, EBV-positive BL and EBV-negative BL [140].

##### Hodgkin Lymphoma

Classical Hodgkin lymphoma (cHL) is a major type of HL, which accounts for approximately 90% of all HLs and is characterized by the presence of scattered malignant HRS cells in the non-neoplastic inflammatory tumor microenvironment background [141]. EBV is present in all of the HRS cells in about 30% of HL cases, the majority of which are the mixed cellularity type [20]. EBV infection of the HRS cells is considered to play a causal role in the oncogenesis of cHL, which is confirmed by in situ hybridization for detection of the EBV genome in HRS cells [20] and EBERs as a target for the routine detection of EBV in cHL [142]. EBV is consistently retained during disease progression, suggesting it is required for maintenance of the tumor phenotype [143]. EBV-infected HRS cells were shown to express a restricted pattern of EBV latency Ⅱ characterized by the presence of EBNA1, LMP1, and LMP2A, as well as a subset of viral miRNA [144]. Furthermore, the HRS cells are germinal center experienced B cells at various stages of development, which lack surface BCR expression due to the crippled rearrangement of BCR [145]. Interestingly, LMP1 and LMP2A are thought to function as surrogates for BCR function, protecting HRS cells from apoptosis, and allow the survival and proliferation of the HRS cells.

##### EBV-Positive Diffuse Large B-Cell Lymphoma

EBV-positive diffuse large B-cell lymphoma (DLBCL), which was formerly designated as EBV-positive DLBCL of the elderly, usually occurs in individuals aged >50 years, with an inferior clinical prognosis in comparison to EBV-negative DLBCL [146]. The incidence of EBV-positive DLBCL is less than 5% in Western countries, but that percentage goes up to 10–15% in Asian and Latin American countries [147]. Most EBV-positive DLBCL cases have an activated B-cell (ABC) phenotype, expressing MUM1/IRF4, and are negative for CD10 and BCL6 [147]. Expression of NF-κB and phosphorylated STAT3 are more commonly seen in EBV-positive DLBCL compared with EBV-negative DLBCL [146]. Gene expression profiles revealed that the activation of the NF–κB and/or JAK/STAT3 pathways are characteristics of EBV-positive DLBCL, even when compared with ABC type EBV-negative DLBCL [148,149]. However, the interaction with EBV is still poorly understood [148]. It has been shown that infection with EBV lacking EBNA3B leads to aggressive, immune-evading, monomorphic, DLBCL-like tumors in NOD/SCID/γc-/-mice with reconstituted human immune system components [150]. There were also additional EBNA3B mutations identified when screening EBV-positive DLBCL human samples [150]. Thus, EBNA3B is a virus-encoded tumor suppressor, and its mutations can result in increased lymphomagenesis.

#### 5.1.2. EBV-Associated NK-Cell and T-Cell Lymphomas

##### Extranodal Natural Killer/T-Cell Lymphoma

Extranodal natural killer (NK)/T-cell lymphoma (ENKTL) is an aggressive lymphoma strongly associated with EBV infection that occurs mainly in Asians and the indigenous populations of Latin American, but rarely in native Europeans and North American populations [151]. The exact role of EBV in ENKTL remains elusive; however, the characteristic geographical distribution suggests that ethnicity plays an important genetic role in regards to the predisposition of these populations to the disease [141,152]. In these cancers, EBV is present as a clonal episomal form with the type Ⅱ latency expression pattern. The EBV type 1 strain is common in ENKTL compared with type 2 [153]. A 30-base pair deletion in the LMP1 gene, C terminus region of the type 2 EBV strain, was common and associated with enhancement of its oncogenic potential in ENKTL [154]. EBV BART9 miRNA modulates LMP1 levels and promotes proliferation of ENKTL cells [155]. Different cytokines (IL-2, IL-9, IL-10, IL-15) are produced by EBV-infected NK cells via autocrine or paracrine loops, and the microenvironment can stimulate cell growth, along with expression of LMP1 [156].

##### Angioimmunoblastic T-Cell Lymphoma

Angioimmunoblastic T-cell lymphoma (AITL) is a neoplasm of mature T-follicular helper (T_FH_) cells strongly associated with EBV infection, which is less common in North America or Asia than in Europe [157]. EBV-positive B cells are detected in more than 90% of biopsies. However, the neoplastic T cells are EBV-negative. To date, the role of EBV in this pathology is still unknown. It has been reported that EBV itself can drive the development of AITL by activating T_FH_ cells [158]. It is also proposed that AITL generates a profound immunodeficiency at the origin of EBV reactivation favoring the expansion of T_FH_ and B cells, thus contributing to the development of the tumor microenvironment (Figure 2). Moreover, the co-expression of *BNLF2a* and *BCRF1* (a homologous protein of human interleukin 10, vIL-10) can contribute to the immune escape and survival of infected cells [159].

### 5.2. EBV-Associated Epithelial Carcinomas

#### 5.2.1. Nasopharyngeal Carcinoma

Undifferentiated (also called anaplastic) nasopharyngeal carcinoma (NPC) has an exceptionally high incidence in the Cantonese population in Southern China (95%), as well as Southeast Asia, Northern Africa, and Eskimos in the Arctic [160]. NPC occurs less commonly in the United States and Europe, with rates of 0.5~2/100,000, about 50-fold less than that seen in Southern China. Undifferentiated NPC virtually always has EBV present in the transformed malignant epithelial cell component, but not in the characteristically infiltrating, substantial population of lymphocytes, which are thought to be important in maintaining cancer by providing cytokines or other signals, and in immune escape.

Besides EBV infection, other risk factors for NPC include male sex, HLA type, and environmental/lifestyle factors [161]. EBV is clonal in high-grade preinvasive lesions (carcinoma in situ or dysplasia), as well as in NPC [162], but not in low-grade preinvasive lesions. Notably, epithelial cells are not as susceptible to transformation as B lymphocytes in vitro, either by co-culture or by transfer infection of EBV from lymphocytes. Only a few NPC cell lines have been generated for use in in vitro studies [163,164]. NPC is believed to be a multistep process by accumulating mutations (overexpression of oncogenes and loss of expression of the suppressor genes), which results in a hyperplastic lesion. Meanwhile, EBV is proposed to act as a trigger by inducing dysplastic lesions and eventually invasive NPC. NPC is described as EBV latency Ⅱ program; however, the expression of LMP1 is often undetectable in many of the cells [165,166]. LMP2A can block STAT3 activation by inhibiting NF-κB, thereby impairing LMP1 expression, which provides a mechanistic basis for the scenario that NPC cells express LMP2A, but very little LMP1 [167]. Fingerprints of Epstein–Barr virus in NPC have revealed the activation of distinct pathways (for example, chromatin modification, ERBB–PI3K signaling, and autophagy machinery) that drive the development of this tumor [168].

#### 5.2.2. Gastric Cancer

EBV was first detected in typical gastric adenocarcinoma tumor cells in 1992 [24]. About 10% of cases of gastric cancer (GC) were confirmed to be EBV-associated GC (EBV-GC), which manifests unique genomic aberrations, significant clinicopathological features, and a good prognosis compared to EBV-negative GC [169]. EBV-GCs also display viral latency Ⅱ program, similar to that of NPC [56]. However, LMP1 is absent in some cases, while BART miRNAs are highly expressed [116]. The precise contribution of EBV to the development of EBV-GC is unknown, but it is shown that EBV-GC development is also linked to several other pathogenic factors, including genetic susceptible background, dietary habits, and metal dust environmental cofactors exposure [170]. The lack of p53 mutation in EBV-GC has led to the speculation that EBV can in some way bypass the need for p53 mutation, which seems essential for most types of carcinomas [9]. Other notable genetic differences between EBV-positive GC and EBV-negative GC include high CpG methylation of host genes upon EBV infection, inactivation of tumor suppressor genes, loss of cyclin-dependent kinase inhibitor p16^INK4A^ expression, and relatively frequent PI3K mutations, to name a few [9,38].

## 6. Summary and Future Perspectives

Since EBV was discovered in 1964, our understanding of EBV has continuously grown. EBV has now been accepted, with its current leading role as a prime example of a human tumor virus that is etiologically linked to an unexpectedly diverse range of diseases, from benign infectious diseases, SADs, to EBV-associated malignancies. Studying the role of EBV in cancer pathogenesis and the host cell–virus interaction, in both humans and animals, has provided principle insights into the mechanisms that drive the oncogenic process. This understanding has also raised the possibility for therapeutic and prophylactic intervention targeting EBV latent/lytic proteins, EBV miRNAs, and the tumor microenvironment, which includes the local cytokine milieu.

There are still many questions that cannot be answered at the moment. Although EBV remains the most common persistent, asymptomatic virus infection in humans, it is unclear why only a small fraction of people develop the viral-associated diseases/malignancies. There are also more studies needed to elucidate the oncogenic contributions to specific EBV lytic proteins in the context of latent infections in certain cancers. The precise role of EBV in the pathogenesis of epithelial cancers, as well as the complex interplay between EBV and the immune system, require additional investigations. However, the challenge would be to exploit these new mechanistic insights to gain a more comprehensive understanding of the biology of EBV infection in vivo and to develop novel therapies for treating virus-associated diseases.

## Figures and Tables

**Figure 1 viruses-15-00714-f001:**
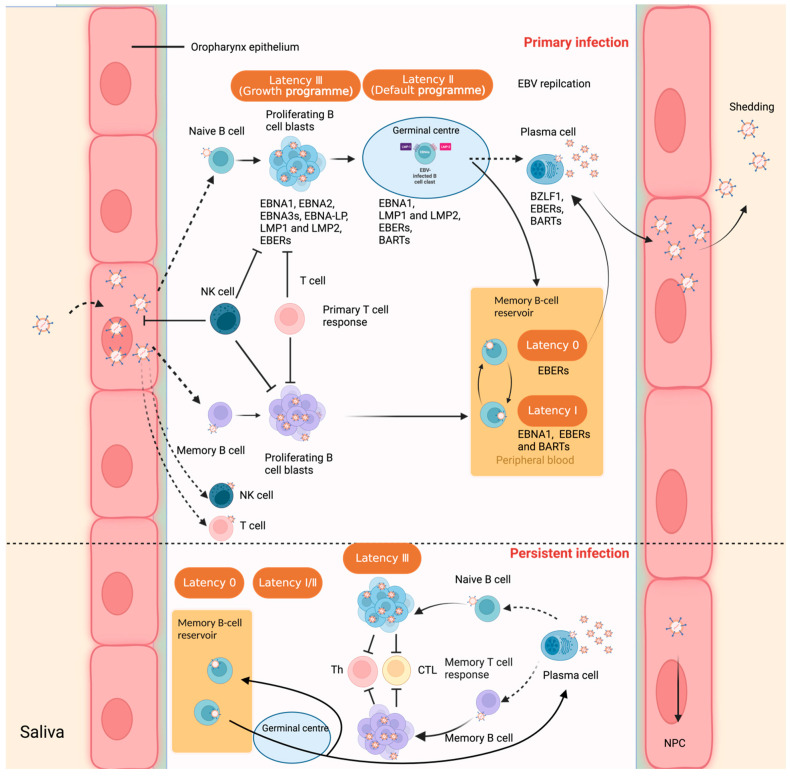
Interaction between Epstein–Barr virus and the human host, and virus latent infection in B lymphocytes. Upper panel: Primary infection. Epstein–Barr virus transmitted via saliva establishes a primary lytic replication in the oropharyngeal mucosal epithelium, then the virus spreads throughout the lymphoid tissues in Waldeyer’s ring. There are two putative theories to explain how the virus enters memory B cells. One speculation proposes that EBV infects tonsillar naive B cells, leading to a latency 3 program; in this scenario, a full spectrum of latent proteins is expressed. The majority of these proliferating cells are eliminated by natural killer cells and the emerging latent-antigen-specific primary-T-cell response. However, some infected cells escape from immune surveillance by downregulating antigen expression and undergo germinal center (GC) reaction, where a more limited set of viral genes are expressed (the default program or latency 2). A stable reservoir of resting viral-genome positive memory B cells is established when these EBV-infected GC B cells migrate to peripheral blood, where viral antigen expression is silenced (latency 0). When EBNA1 is expressed intermittently during the division of these memory B cells, the viral genome is distributed to the daughter memory B cells (latency 1). Another view envisages that EBV directly infects pre-existing memory B cells in the memory B-cell reservoir. Memory B cells can terminally differentiate into plasma cells (solid arrow), possibly moving to the oropharyngeal mucosal basolateral side and, in the process, triggering the viral lytic replication. Virions produced at these sites are efficiently shed into the saliva and transmitted both to other hosts and to previously uninfected naive B cells within the same host. EBV-infected GC B cells might also differentiate directly into plasma cells (dashed arrow). It is also reported that EBV can also infect T cells, NK cells to form T-cell leukemia/lymphoma, NK-cell leukemia, and NK/T-cell lymphoma [37,38]. Lower panel: Persistent infection. The reservoir of EBV-infected memory B cells are normally in a dormant status. Under certain circumstances, these cells might be recruited into GC reactions, after which they might either re-enter the reservoir as silent memory B cells or return to the lymphoid tissue and undergo plasma cell differentiation, shedding EBV virions. This may initiate the growth-transforming latency III program on infection of naive and/or memory B cells. These new infections are more likely to be efficiently removed by the well-established memory-T-cell response. This figure was created with BioRender.com (accessed on 5 December 2022).

**Figure 2 viruses-15-00714-f002:**
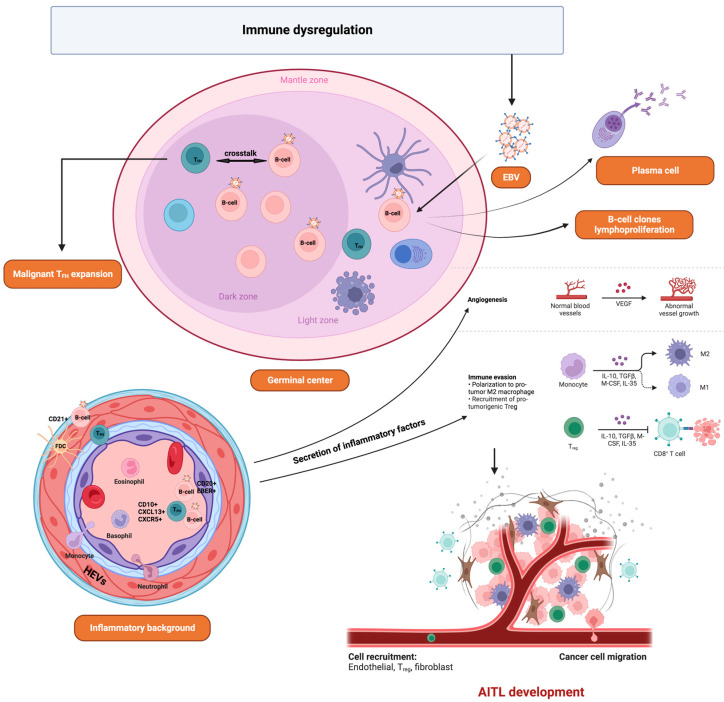
Pathogenetic model of AITL. In AITL, a complex network of interactions take place between the tumor cells and profound surrounding inflammatory tumor microenvironment. FDC, follicular dendritic cell; HEV, high endothelial venule; T_FH_, follicular helper T cell; T_reg_, regulatory T cell.

**Table 1 viruses-15-00714-t001:** EBV viral gene expression patterns during different types of latent infection.

	Latency Pattern	Latency III	Latency II	Latency I	Latency 0
Genes					
EBNA1		+	+	+	ND
EBNA2		+	ND	ND	ND
EBNA3s		+	ND	ND	ND
EBNA-LP		+	ND	ND	ND
LMP1		+	+	ND	ND
LMP2		+	+	ND	ND
EBERs		+	+	+	+
BHRF1 miRNAs		+	ND	ND	ND
BARTs miRNAs		+	+	+	+

ND: not detected. +: positive.

## Data Availability

Not applicable.

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
