# Peer review of "Epstein–Barr Virus History and Pathogenesis"

_viruses, 2023, doi:10.3390/v15030714_

Round 1
Reviewer 1 Report
No comments or suggestions to improve.
Author Response
Thank you for your acknowledgement to our work.

Reviewer 2 Report
Epstein-Barr virus (EBV) was the first human tumorigenic virus to be characterized. In the sixty years since its discovery, scientists from around the world have made remarkable contributions to our understanding of various aspects of biology and the development of new technologies in the field of EBV virology. For instance, de novo EBV infection transforms primary B lymphocytes isolated from human peripheral blood mononuclear cells, providing a perfect platform to investigate oncogenesis, viral entry, and gene expression regulation. EBV has tropism for a variety of host cell types, encodes a large set of viral oncogenes, and is associated with multiple human malignancies, making it possible to study unique mechanisms underlying oncogenesis in different types of cancer. EBV in Burkitt lymphomas undergoes tight virus latency, and viral reactivation can be achieved through chemical triggers, providing an excellent model for researching the mechanisms of herpesvirus maintenance and regulation of herpesvirus lytic replication. As our understanding of EBV virology deepens, research findings will benefit other fields as well.
This manuscript by Yu et al. provides a brief review of the history of EBV discovery, expertly depicts EBV life cycles, summarizes viral genes with oncogenic potential, and outlines the different malignancies associated with EBV infection. The paper is well-written and organized in a logical format, providing an informative summary of the discoveries in the EBV field with sufficient references. However, a few concerns need to be addressed to improve the manuscript.
Major comments:
1. Although this manuscript itself stands in as a very informative article, this reviewer found it kind of lack of illustrations. More schematic figures describing the roles of EBV oncogenes in different types of malignancies are highly encouraged to be included.
2. Writing is dense and sometime redundant. For example, there are too many ‘and’ used and same word was used too frequently within a single paragraph (one example is the repeated appearance of ‘essential’ between line 189 and 190). Editing is needed to further streamline the text body.
Minor comments:
1. Between line 64 to 68, why NPC was not listed among malignancies EBV DNA was
detected in?
2. Line 88, EBV shows strong tropisms to B and epithelial cells not the other way around stated.
3. What does ‘chronic relapse’ mean in line 135? A more precise term should be replaced with.
4. It seemed to be self-contradicting within the text between line 361 and 363 about the availability of NPC cell lines. Please be aware of the presence of C666-1 and recently established NPC PDX cell lines.
5. The term ‘high lytic infection’ in line 253 is not a scientific term.
Author Response
Thank you for your reminder. We have corrected the issued you pointed out one by one.
